# Asymmetries of Forelimb Digits of Young Cattle

**DOI:** 10.3390/vetsci7030083

**Published:** 2020-07-02

**Authors:** Pere M. Parés-Casanova, Laura Castel-Mas, Kirian N. Jones-Capdevila

**Affiliations:** Department of Animal Science, ETSEA, University of Lleida, Av. Rovira Roure 191, 25198 Lleida, Catalonia, Spain; lcm13@alumnes.udl.cat (L.C.-M.); kjc1@alumnes.udl.cat (K.N.J.-C.)

**Keywords:** digital bones, directional asymmetry, laterality, locomotion, Pyrenean Brown

## Abstract

Based on the anatomical premise that, in bovines, the medial (inner) hoof is larger than the lateral (outer) one in the forelimb, we hypothesized that this implies a phalangeal form difference. To test this hypothesis, asymmetries of the forelimb acropodia (phalangeal series) were studied on calves belonging to the Brown Pyrenean breed, a meat breed managed under extensive conditions in NE Spain. Dorso-palmar radiographs were obtained for each acropodium in a sample of 17 paired left and right forelimbs. Size and shape were analysed by means of geometric morphometrics on medial and lateral acropodial series (III and IV series respectively) for each left and right limb. Shape coordinates were computed by Generalized Procrustes Analysis. Medial and lateral acropodial series appeared similar in size, but their shape expressed an especially high directional asymmetry, with distal phalanges (pedal bones) being abaxially (outwards) oriented. Such morphological observations may be an important reconsideration of “normal” radiographic acropodial symmetry evaluation. This can be explained not only by an unevenly distributed ground reaction force between acropodial series, but also between right and left limbs, making medial and lateral hoof surfaces differently prone to overloading and, accordingly, to injuries to the limb.

## 1. Introduction

In bilaterally symmetric structures, size and shape can differ between sides [1,2]. Bilateral asymmetries can be expressed as fluctuating asymmetry, directional asymmetry and antisymmetry [3,4,5]. Fluctuating asymmetry is generally the result of genetic or environmental stress [6,7,8,9] and is considered a negative indicator of the ability to resist random and small developmental accidents [3,10]. Directional asymmetry occurs when there is a consistent form difference between sides [5,11,12] and generally it has a genetic basis [13], usually not being considered to be due to environmental or developmental stress [12], although there is evidence of transitions from FA to both DA and antisymmetry under stress [14]. In antisymmetry, there is a departure from a Gaussian distribution of right minus left differences (bimodal distribution) [15].

The cattle metacarpus is composed of the two totally fused third and fourth metacarpal bones [16,17]. This composite bone, colloquially known as the “shinbone” [18], provides two separate epiphyses for the third and fourth digital series [16,17]. Although cattle have four digital series, the abaxial (II and V) are vestigial and incomplete [16,17], whereas the axial series (III and IV) are composed of three primary (distal—the “pedal bone”, middle—the “short pastern”, and proximal—the “long pastern” phalanges) and a sesamoid bone (“navicular bone”), jointly forming the acropodia [18,19]. The proximal and middle phalanges are outside of the hoof and the distal phalanx and navicular bone are inside the hoof [18].

Like most bone skeletal structures, acropodial series III and IV are apparently bilaterally symmetrical [20], and existing published papers speak to their asymmetry [21,22], describing an asymmetry of the digits, especially with respect to their length, being lateral phalanges longer than their medial counterparts. The lateral (outer, IV) vs. medial (inner, III) symmetry of the acropodial series in the same limb corresponds to object symmetry, e.g., III and IV are identical according to an axial plane [23]. The matched symmetry, in contrast, refers to separated series, which appear as mirror images of each other [23], as would occur between the right and left limbs.

The use of traditional (linear) morphometrics in studies of bilateral asymmetry has some limitations [24]. For instance, linear distances omit information about relationships involving their shape [24]. Geometric morphometrics (GM) are derived from a set of landmarks based on Cartesian coordinate points (*x*,*y*) that are homologous across individuals [25]. In contrast to traditional linear methods, this method preserves the geometry of the measured landmark configurations, and statistical results can thus be represented as shape deformations [24,25,26]. Moreover, GM is very helpful in standardizing and separating the different sources of asymmetry [25,27]. Analyses of asymmetries in appendicular (such as the autopodes) or in axial skeletal elements are very scarce in cattle [28,29,30,31] and GM has never been applied to the study of acropodium asymmetries in bovines. This study determines asymmetries in acropodial series in calves based on radiographs, as, to our knowledge, radiographic geometric morphometric analyses of digits have not previously been used for cattle. Previous studies have centered on the forelimbs, which take on a lesser role in propulsion and are the major weight-bearing structures [16,32], and so are less prone to change according to substrates [18,33,34]. Moreover, as the chosen sample belonged to animals not subjected to foot care, it is expected that we will describe “natural” acropodial asymmetries, at least among young animals similar to the studied breed.

## 2. Materials and Methods

A sample of 17 pairs (right and left) of forelimbs from 17 Pyrenean Brown calves (9–15 months and 212–282 kg carcass weight as standard values) [35] was collected in a commercial slaughterhouse. The animals were clinically sound according to ante mortem abattoir official veterinary inspection and with no previous trimming. Individual information was not possible for samples, therefore sex and carcass weight for each animal could not be considered, although there were no castrated animals. The limbs were identified as the feet came off the slaughter line. Each set of paired feet for each animal was placed in a uniquely identified plastic bag, brought to the Department of Animal Science and stored at −18 °C until processing.

Images were obtained using a computed radiographic system (Potro DR^®^) following standard procedures [36]. The exposure factors for dorso-palmar view were 60 kV and 3.2 mAs, with each defrosted foot to be examined with the X-ray beam centered approximately on the metacarpophalangeal joint (Figure 1). The cassette was vertically positioned as close as possible to the metacarpus. A total of six landmarks occurring on both sides (axial and abaxial) of the acropodial joints, one on the abaxial metacarpal distal joint and one on the axial distal metacarpal epiphysis, were used (Figure 1). These landmarks were chosen in order to give a good representation of the overall acropodial shape and to detect important features of asymmetry. The captured images were transformed to tps files using TpsUtil v. 1.40 software [37] and landmarks were recorded using TpsDig v. 2.26 software [38].

To obtain the shape data, landmark configurations were superimposed using the generalized Procrustes method, based on a generalized least-squares minimization of the distance between corresponding landmarks [39]. Landmark configurations were compared by this superimposition, which is achieved by translating, rotating and scaling all configurations to a common reference system (the mean) [27]. As a proxy for size, we used the centroid size (CS), which corresponds to the squared root of the sum of the squared distances from each landmark to the centroid [39].

### Statistical Analyses

Size differences were analyzed with an Analysis of Variance (ANOVA) of the log centroid sizes. Procrustes coordinates were assessed for asymmetry of shape in relation to asymmetric components of variation (Directional Asymmetry (DA) and Fluctuating Asymmetry (FA)), and measurement errors. FA was interpreted as the “individual*side” interaction effect, while DA was interpreted as the “side” effect. This was done with a two-way Non-Parametric Multivariate Analysis of Variance (NPMANOVA), using “individuals” and “sides” as factors. Allometry was studied by means of a multivariate regression using CS (log transformed) as an independent variable and shapes (expressed as Procrustes coordinates) as dependent variables. A subsequent shape analysis was done by Principal Component Analysis (PCA) to simplify descriptions of differences between the series. PCA was calculated from the covar matrix of the regression scores. Finally, a representation of bending energy (thin plate spline) allowed us to appreciate shape changes as forces acting on a coloured mesh that deform according to the vector direction.

The obtention of size and shape data was done with MorphoJ v. 1.07a [40] and statistical analyses were done with PAST v. 2.17c [41].

## 3. Results

### 3.1. Size Asymmetries

ANOVA reflected that III and IV acropodial series presented no size differences between them for both limbs (*F*_3,132_ = 0.156; *p* = 0.925), irrespective of limbs.

### 3.2. Allometry

There appeared to be no significant regression of CS vs. Procrustes coordinates (F_16, 119_ = 0.719, R^2^ = 0.0076, *p* = 0.769), so there was no allometric effect of acropodial size vs. shape.

### 3.3. Shape Asymmetries

Procrustes ANOVA, for assessing measurement errors, showed that the mean square for interaction effect exceeded the measurement error; thus, the measurements were deemed reliable. There appeared to be directional asymmetries (DA, “side” effect) and fluctuating asymmetries (FA, interaction effect) in each right and left forelimb. Net asymmetry (DA + FA) accounted for more than 90% of the total variation, with DA being more than twenty times bigger than the value of FA (Table 1). In the PCA, the first two principal components represented 46.80% of the total shape variability (Figure 2), with Principal Component 1 (PC1) accounting for the most significant variance, 31.04% (Figure 3). The primary shape characteristic of the dataset, as represented by the first PC, may be interpreted as a horizontal displacement of distal landmarks—landmarks 3, 4, 7 and 8 on the distal and middle phalanges showed the highest contribution to variance—with a more marked displacement towards the right (Figure 4).

In the representation of bending energy (contraction, bluish coloration and expansion, reddish coloration) we observed a left (abaxially) displacement of distal phalanges in left lateral and right medial series, while this displacement was towards the right (also abaxially) for the right lateral and left medial series. (Figure 5). In other words, we found an abaxial displacement for paired pedal bones, which was more marked towards the right.

## 4. Discussion

In this study, we analysed acropodial asymmetries in young domestic cattle using radiographic images by means of geometric morphometric techniques. The undeniable presence of acropodial shape asymmetry in the studied sample appeared. In cattle, the medial hoof being the larger hoof on the forelimb has been previously described [16,32,42]. According to our results, this size difference is not reflected in the bones, as acropodial sizes are similar between the medial and lateral series. However, this asymmetry was indeed present in their shape. Moreover, once the net asymmetry of the studied sample was decomposed into directional asymmetry and fluctuating asymmetry, it was found that most of the variation was explained by directional asymmetry alone. Directional asymmetry refers to non-pathological differences between sides, which result from lateralized behaviours or biomechanical pressures [24].

The directional asymmetry can be a product of genotype as well as of lateralization in the muscle load or other functional demands [43]. The most probable explanation for the observed directional asymmetry in acropodial bones is that it results from a functional splay due to an abaxial displacement of two paired pedal bones, in which the phalanges tend to fall outside the main vertical axis supporting the body weight [44]. In fact, it is described that hooves splay because the abaxial wall is longer than the axial wall [18]. Externally, this can be seen in the “open” oriented apexes of the hooves [45,46]. The novelty of our research is a right-biased detected directional asymmetry, which would imply uneven weight distribution between the medial and lateral hooves.

In summary, our results showed significant differences in the paired shape—but not size—of the medial and lateral distal acropodia, with a clear abaxial-biased asymmetry that was more marked towards the right. It would be interesting to study this expression in more aged animals, and to detect whether this directional unevenness coincides with both kinetic and kinematic asymmetrical locomotor differences. The hypothesis that the ground reaction force is differently and unevenly distributed between hooves for right and left limbs would make hoof surfaces differently prone to overloading and injuries to the limb.

## Figures and Tables

**Figure 1 vetsci-07-00083-f001:**
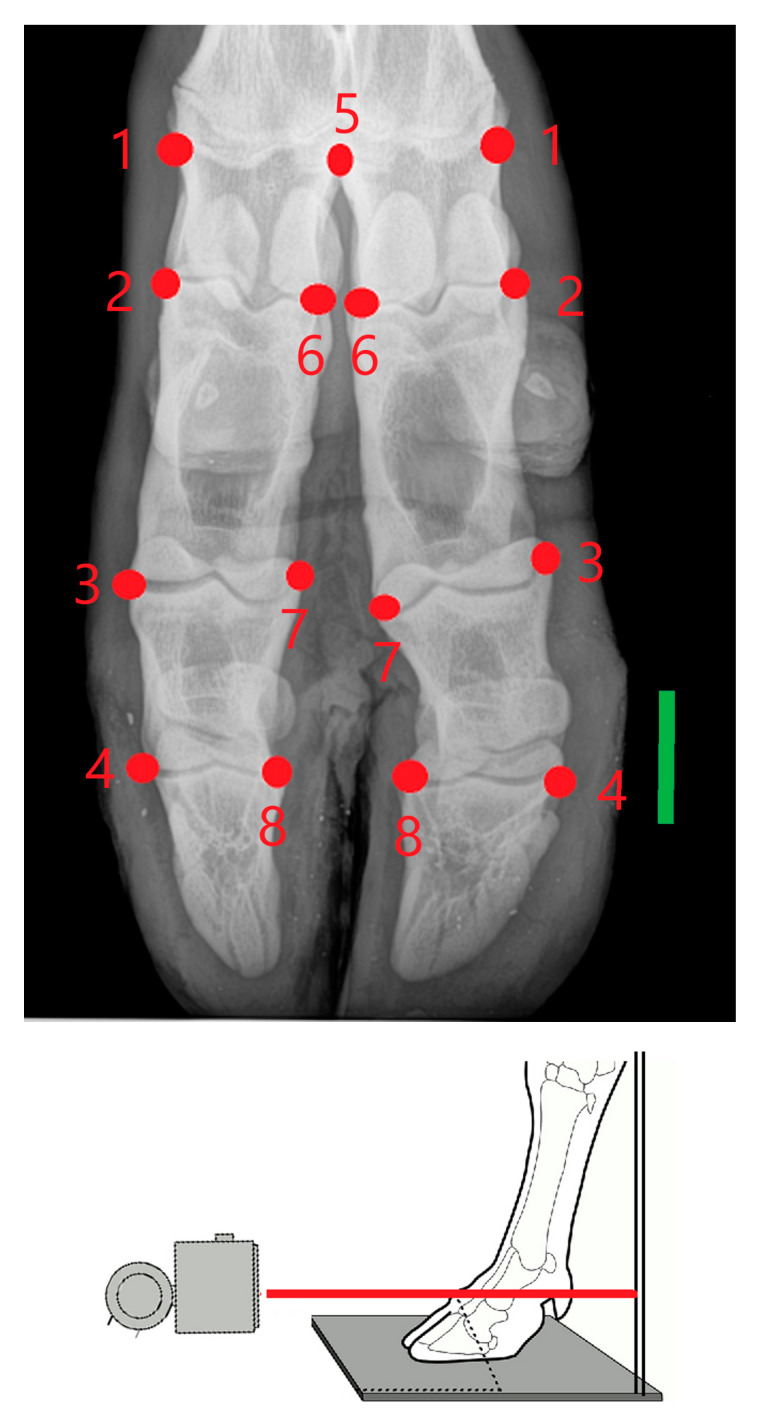
Dorso-palmar view of acropodium on which seven landmarks occurring on both sides of the acropodium joints (no. 2 to 4, and 6 to 8), abaxial metacarpal distal joint (no. 1) and one axial (no. 5, on the distal metacarpal epiphysis) were located. The bar is 3-cm long. For each limb, the X-ray beam in the images passed through the proximal interphalangeal joint.

**Figure 2 vetsci-07-00083-f002:**
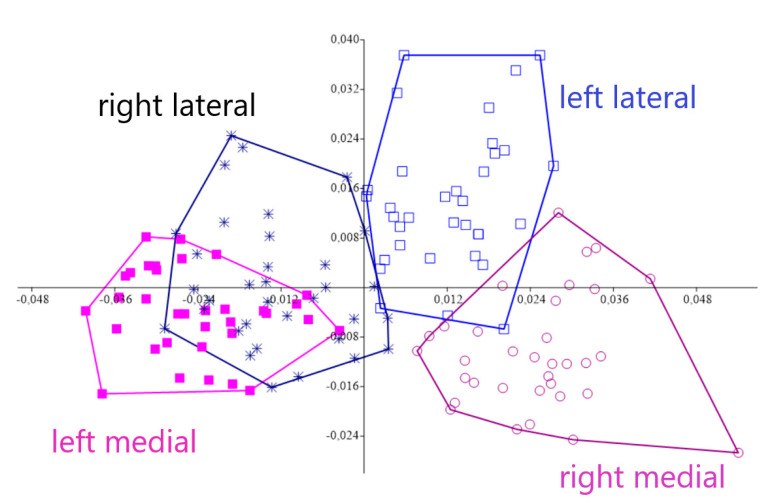
Principal Component Analysis for left and right acropodium asymmetries belonging to 17 Pyrenean Brown calves. The first two Principal Components were statistically meaningful and portrayed 46.80% of the total shape asymmetry, with PC1 accounting for 31.04% of the variance.

**Figure 3 vetsci-07-00083-f003:**
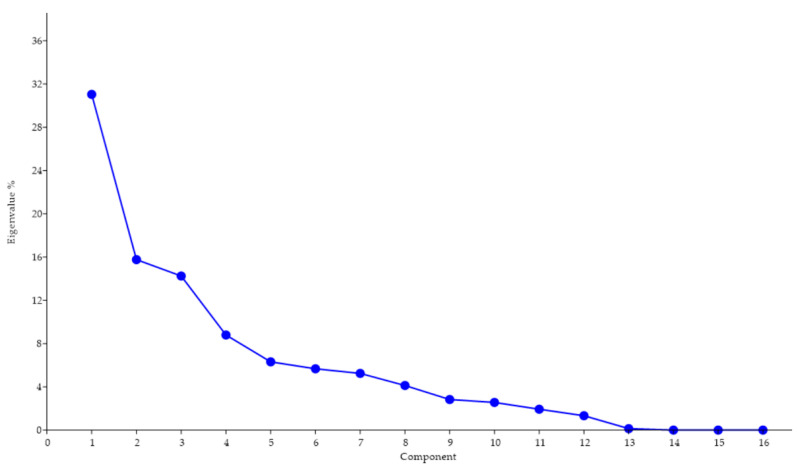
Screen plot showing the proportion of the total shape variance explained by each PC for 17 pairs of distal forelimbs belonging to Pyrenean Brown calves. There was a gradual decrease in the percentage of variance. The first two PCs accounted for 46.80% of the total variation.

**Figure 4 vetsci-07-00083-f004:**
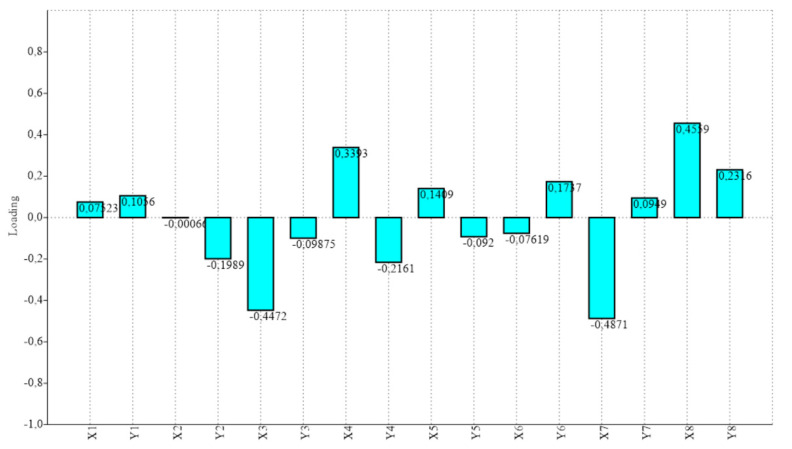
Loadings for the first PC, which accounted for 76.17% of the total observed variance for each of the eight studied landmarks (expressed in *x*, *y* Cartesian coordinates). Distal landmarks—on distal and middle phalanges, landmarks 3, 4, 7 and 8—showed the highest contribution to variance, with a horizontal (*x*) displacement tendency.

**Figure 5 vetsci-07-00083-f005:**
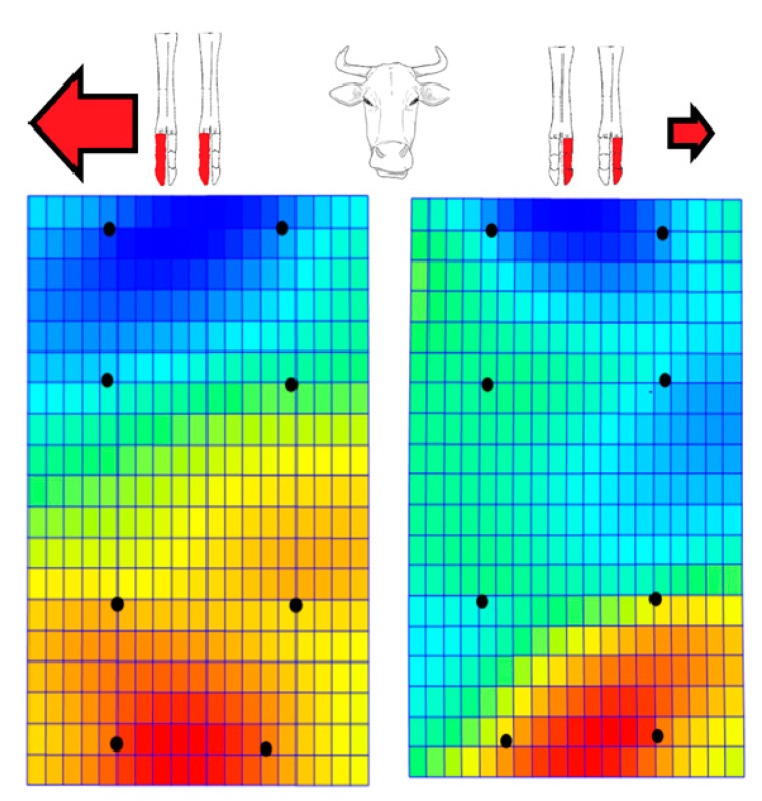
Representation of bending energy (contraction, bluish coloration and expansion, reddish coloration) for 17 paired limbs from 17 Pyrenean Brown calves (dorsal views). The right lateral series and left medial series on the left (above) appear on the left net frame (down) and the left lateral and right medial series (above) appear on the right net frame (down). The black dots equate to the previous landmarks (see Figure 1). We observed displacements of distal phalanges with a tendency towards abaxial displacements for paired acropodial series, which were more marked towards the right.

**Table 1 vetsci-07-00083-t001:** Procrustes ANOVA for shape of right and left paired forelimbs of Pyrenean Brown calves (n = 17), with a significant “Side” (directional asymmetry) and interaction effect (fluctuating asymmetry, 9999 permutation rounds). Sums of squares and mean squares are in units of Procrustes distances (dimensionless). The measurement errors showed that mean squares for interaction effects exceeded the measurement errors. There appeared to be directional asymmetries and fluctuating asymmetries for both right and left forelimbs. Sums of squares and mean squares are in units of Procrustes distances (dimensionless).

Source	Sum of Squares	Df	Mean Squares	*F*	*p*
Right forelimb (n = 17)
Individual	0.032029	16	0.0020018	32.306	0.0001
Side	0.027964	1	0.0279640	45.128	0.0001
Interaction	0.018533	16	0.0011583	18.693	0.0002
Residual	0.021068	34	0.0006196		
Total	0.099594	67			
Left forelimb (n = 17)
Individual	0.033277	16	0.0020798	40.677	0.0001
Side	0.028572	1	0.0285720	55.881	0.0001
Interaction	0.019052	16	0.0011908	23.289	0.0001
Residual	0.017384	34	0.0005113		
Total	0.098285	67

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
