# Peer review of "Asymmetries of Forelimb Digits of Young Cattle"

_vetsci, 2020, doi:10.3390/vetsci7030083_

Round 1
Reviewer 1 Report
Line 10 use passive voice and delete al “we” example: ayymetries of the forelimb acropodia were studied, like line 12.
Line 11 delete form, this concept is not used correctly for the kind of analysis you are doing the correct word is shape and delete (size+shape) this sugerence was made in the first revision and was not taking into account.
Line 24 [3] [4] [5] is not the correct way to add citation if they are consecutive number should be [2-5] please read and follow the MDPI correct information to authors this is also not the first time we recommend to the author to correct the citation in the correct form.
Line 71-72 Did only one landmark explain variation in an anatomical section?
Line 90: use passive voice a recommendation is did not use TpsSmall can be avoided only calculating the correct approximation of 2D 2k-4, the use of this software is indeed very old fashioned and have to be avoided .
Line 92-93 How did you perform the ANOVA (check fonts), did you export the centroid size? And perform it with some software? Please define it.
Line 97. I did not understand why you perform a NPMANOVA when MorphoJ have the option to calculate the FA and DA by a Procrustes ANOVA implemented in the Software and I guess is even a better way to calculate it…
Please be clear which analysis you perform with MorphoJ and which one you did in Past is not clear.
Line 102 – wrong the PCA can not be calculated from a var-cov matrix, because there are differences between them… is only calculated by a COVARIANZE matrix. Which include negative values.
Line 103- No need to repeat what you already said in the previous paragraph.
Line 117-118 - Why you get a R2 are you performed a Linear Regression? Check if you perform correctly a multivariate regression…
Line 125- You mentioned “In the PCA the first two principal components represented a 92.43% of 125 total shape variability, with PC1 accounting for the most significant variance (76.17%)” and there is no allometric effect?, I have my doubts with this results if you perform correctly the allometric regression in MorphoJ I think some of this % should be attributable to allometry, please explain here why not.
Line 131: “CV1 explained 98.6% of variation in the analyzed 131 material and it clearly separated each group” - Please delelete the % explained by the CVA CVA only maximize the variance from the values obtained from the PCA is wrong explain the % of a discriminant analysis have no sense at all.
Also I don’t think the necessity to include the CVA when the PCA explain very well the differentiation in shape, I must also consider exclude the CVA from the analysis.
Figure 3 did not explain very much to be a figure itself, I recommend delete it and explain it only in results.
Author Response
Line 10: passive voice used.
Line 11: deleted word "form".
Line 24 and others: sequenced citations arranged.
Line 71-72: only one axial landmark, in fact.
Line 90: passive voice used. We have not deleted this test. Must I delete this test?
Line 92-93: "ANOVA (ANalysis Of VAriance) of log centroid sizes", added.
Line 97: yes, we performed a NPMANOVA with PAST software, not MorphoJ. Significance values are identical for both (I have just corroborated it again). SS and MS absolute values are different between softwares, but not their relative contribution. We have clarified applications we did for each software (line 107). See Anderson, 2001. A new method for non-parametric multivariate analysis of variance. Austral Ecology 26:32-46.
Moreover PAST software allows much illustrative draws, and this is why we prefer it, too.
Line 102: corrected. I did not know it!
Line 103: second sentence deleted.
Line 117-118: the value R2 appears as an overall stats as it was another reviewer's suggestion
Line 125: values corrected. It was a numerical fault (we had applied a "between-group" option and it had to be "disregard", uops!). Loadings and most discriminative values are the same, obviously.
Line 131: % deleted. In our opinion, CVA reinforces the idea the ordination of 4 studied groups. Anyway, if necessary, we will delete the test and its associated figure.
Figure 3: we think that this figure adds more clearity to the main idea than did the former one in first draft. Again, if considered unnecessary, we wil delete or change it.
The manuscript has been modified in deep according to 4 reviewers' comments.
Reviewer 2 Report
The data and its presentation in the manuscript are much improved since the initial submission. I think this paper will make a good addition to the research literature.
Author Response
Thanks. The manuscript has been modified in deep according to 4 reviewers' comments.
Reviewer 3 Report
8 - reference that speaks to the anatomical premise described
12 - radiographs
18 - This finding would not surprise someone regularly evaluating bovine radiographs. This paper is just describing another way of evaluating the orientation.
36-38 There are published papers which speak to asymmetry so I don't think it’s correct to say that symmetry is ‘generally assumed’.
Keller, A., Clauss, M., Muggli, E., Nuss, K. Even-toed but uneven in length: the digits of artiodactyls. (2009) Zoology, 112 (4), pp. 270-278. doi: 10.1016/j.zool.2008.11.001
Muggli, E., Weidmann, E., Kircher, P., Nuss, K. Radiographic Measurement of Hindlimb Digit Length in Standing Heifers. (2016) Journal of Veterinary Medicine Series C: Anatomia Histologia Embryologia, 45 (6), pp. 463-468. doi: 10.1111/ahe.12222
49 - reword is no equal is not good English
52 - This study
54 - ref to support the statement regarding substrates
60 - The word random has a specific meaning when it comes to study design. This means you need to describe the randomisation process. If you collected limbs from a set of animals going through the abattoir based on those being the ones you had permission for and when it was convenient to do so during meat processing you can't call it random.
64 – 2à two or even just each set of paired forelimbs...
82 (fig 1 legend) states beam centred on the metacarpophalangeal join, beam in image passes through the proximal interphalangeal joint.
153 distal forelimbs
169 – fig 6 legend define the black dots – do these equate to the previous landmarks?
175-176 it appeared… reword this sentence it doesn’t read correctly
179 – make sure that what “this” is very clear to the reader – its currently confusing… probably because previous section doesn’t read well.
187 no need to put pedal bone again – already defined.
end 187 -193 – The meaning of this section is not at all clear
193 refs 36 and 38. How are these references relevant when there is no radiography involved in the studies?
197 re word this sentence does not read correctly
Author Response
Line 8: I do not understand.
Line 12: changed for "radiographs".
Line 18: we have not found references taking about this bilateral asymmetry ("right and outwards and left hand inwards", or in other words, a right supination accompanied by a left pronation).
Lines 36-38: both references added and sentence changed.
Line 49: rewritten.
Line 52: corrected.
Line 54: a reference has been added.
Line 60: "randomly" has been supressed, as in fact it added nothing for ulterior studies.
Line 64: added "from the same animal".
Line 82 (fig 1 legend): changed to "beam in images passes through the proximal interphalangeal join".
Line 153: changed to "distal forelimbs".
Line 169: added "Dots equate to the previous landmarks".
Lines 175-176: changed to "But this size difference is not reflected on bones".
Lines 177-179: new sentence: "But this asymmetry was indeed present in shape. And once net asymmetry of the studied sample was decomposed into directional asymmetry and fluctuating asymmetry, it was found that most of variation was explained by directional asymmetry alone".
Line 187: "pedal bone" has been deleted.
Lines 186-188: new sentence: "Detected directional asymmetry can be largely attributable to differential mechanical loading due to claw splay, in which the phalanges tend to fall out the main vertical axis supporting the body weight".
Lines 191-192: new sentence: "In summary, our results showed significant differences in the paired shape -but not size- of the medial and lateral distal acropodia with a clear abaxial-biased asymmetry".
Lie 189: new sentence: "Externally it can be seen as an abaxially oriented of medial and lateral apexes of the hoof".
Lines 192-194: new sentence: "Now it would be interesting to study if this asymmetry has a sexually dimorphism expression and its behaviour in more aged animals".
Line 197: "obtained" instead of "performed".
The manuscript has been modified in deep according to 4 reviewers' comments.
Reviewer 4 Report
The authors describe shape asymmetry in the forelimbs of Pyrenean cattle. The paper is brief and relatively well written, though there are some awkward sentences.
Contrary to comments in the paper, transitions from directional asymmetry (or antisymmetry) are common in the asymmetry literature. Moreover, directional asymmetry is only slightly more heritable than fluctuating asymmetry. See the many papers by Larry Leamy on fluctuating and directional asymmetry in mice. There are also several published papers on disentangling directional and fluctuating asymmetry.
A figure showing the labeled configuration of the limb would be nice. A biologist who doesn’t study ungulates may have a hard time understanding some of the Introduction (2nd and 3rd paragraphs).
With respect to sampling, right and left forelimbs on the same individual are not independent of one another. Consequently, the sample size is n = 17, not n = 34. Doing so constitutes pseudoreplication. The authors apparently understand this because they analyze left and right limbs separately. Mentioning why they did this is important. In addition, pooling right and left sides for the PCA is fine, because you are not engaging in inferential statistics.
Please estimate the variance components of the individuals, sides, individual x sides interaction, and measurement error in the ANOVAs. The wording of the statement that measurement error mean square should not exceed the individual x sides mean square implies a misunderstanding of ANOVA and inappropriately suggests a comparison of variance components when all you are comparing is mean squares. Mean squares are not variance components (with the exception of the error mean square). The individual x sides interaction is the sum of weighted Var(I x S) and Var(measurement error). Var(error) can easily exceed Var(I x S) when Var(I x S) > 0. That is all you need care about, that Var(I x S) > 0.
I’m not sure that you really need both a PCA and a Canonical Variate Analysis. I would just go with the PCA, which reflects the natural variation. The CVA maximizes the differences, but those differences are still evident in the PCA.
The Discussion can be expanded. Two paragraphs are entirely too brief.
I’ve included several comments on the manuscript itself, in pdf format.

Author Response
CVA has been deleted and last figure has been improved (in our opinion). Loading values tables are more visual that a mere table, but we can change it if necessary. The same for screen plot (moreover it illustrates quite well the lack of allometry).
The manuscript has been modified in deep according to 4 reviewers' comments.
Round 2
Reviewer 1 Report
The authors have revised correctly the comments, still I have my opinions about the NPMANOVA dor geometric morphometrics analyses but if they are reffering this with cited data, should be correct for they version. I have no more comments
Author Response
Version has been slightly modified according to Reviewer 3's comments. Changes appear highlighted.

Reviewer 3 Report
Line 18 - any one looking routinely at x-rays would be aware of the conformation so this is not a new consideration.
Line 19 - You can't 'open' a hypothesis which has already got published work related to it - you even cite someof it. You can say if you work supports or refutes previous findings.
Line 40 - by secondary bone do you mean sesamoid bone, or something else? Some extra" marks in this line.
Lines 43-44 - this does not make sense - what are you trying to say?
Line 44 - you could use the content of these papers to illustrate the magnitude of the asymmetry these authors described in their studies...
Line 56 - You cite several sources what do they say about asymmetry? Do they support or refute your argument?
Mats / methods reads better.
Line 77 - doesn'yt aline to Fig 1 legend.
Line 79 - 6 + six
FIg 1 legend the landmark ID is still not clear. what are the bony landmarsks for each of 1-8 ?
Line 114 - irrespective
Line 127- 3 and 7 are not on the distal phalanx. so this needs rewording same issue in fig 4 legend line 152.
Fig 5 - These limb images have got me very confused it looks like you have coloured the medial digit of one limb and lateral digit of the other?
It's hard to know what you mean by 'right' in the bending towards the right summary... Are you treating medial digit from left and right limbs as one group and the lateral digit of left and right limbs as another group - because you demonstrated no difference between the right and left forelimb? If so I think you need to alter which digits are highlighted... Legend also needs attention to make this clear. Line 168 - ref 32 should cite R Blowey in the specific chapter not A Andrews and the whole text book. Consisitency of nomenclature claw vs hoof 181 You results --> support the findings of which previous work?Author Response
Lines 18-19: paragraph changed to "“normal” radiographic acropodial symmetry evaluation."
Line 19: changed to "This can be explained by a ground reaction force unevenly distributed...."
Line 40: "secondary" changed to "sesamoid". Subtle anatomical observation, thanks.
Line 43: added "acropodial series".... In my opinion is it totally well expressed.
Line 44: added a short paragraph explaining cited authors' conclusions: "which describe an asymmetry of the digits, especially with respect to their length, being lateral phalanges longer than their medial counterpart."
Line 56: these references appear just to express the fact that GM has not been applied previously to this kind of researches.
Line 77: I do not understand.
Lines 79-81: corrected to "A total of six landmarks occurring on both sides (axial and abaxial) of the acropodial joints, one on abaxial metacarpal distal joint and one axial on distal metacarpal epiphysis were used (Figure 1).".
Lines 87-89 (correponding to fiure 1 legend): "occurring on both sides of the acropodiums joints (no. 2 to 4, and 6 to 8), abaxial metacarpal distal joint (no. 1) and one axial (no. 5, on the distal metacarpal epiphysis) were located".
Line 115 (ancient line 114): corrected.
Line 128 (ancient line 127): corrected to "on distal and middle phalanges".
Lines 152-153: corrected to "Distal landmarks -on distal and middle phalanges, landmarks 3, 4, 7 and 8-".
Figure 5: added the following sentence: "Right lateral series and left medial series on left (above) appear on left net frame (down) and left lateral and right medial series (above) appear on right on right net frame (down)" (lines 162-164).
Reference 32 changed.
Consistency of nomenclature: "hoof" and "hooves" instead of "claw" and "claws".
This manuscript is a resubmission of an earlier submission. The following is a list of the peer review reports and author responses from that submission.
Round 1
Reviewer 1 Report
vetsci-752344
Asymmetries of forelimb digits of young cattle
The authors have attempted to study the influence of pattern of asymmetry in the forelimb of young cattle, the methodology seems to be challenging for the topic in veterinary sciences, the study of pattern of asymmetry can dilucidated effect on the developmental stability of the organism. The article look interesting for the reader, need include some citation that were recommended to include in the text (pdf attached with comments), so minor revisions be needed to be finally accepted.
I missed in the Methodology the Principal Component analysis, in Geometric Morphometrics one of the main analysis is the PCA in order to represent the shape space. Using the PCA data the authors are explaining the shape explanation to the variation so I must recommend remove the section in results where the authors indicate the CVA % and use the real shape explanation information from the Principal Component Analysis.
Comments on the article can be find in the PDF attached

Author Response
All manuscript has been deeply modified and new references have been added. Conclusions realted to weight bearing and loadings have been deleted and conclusions are now in total agreement with “normal” anatomical hoof constitution. This is why some of Reviewer's comments are not discussed here. PLS analysis has been deleted (as we considered it added nothing to the research) and table 1 has been remodelled. Three figures have been added and the rest have been modified according to general reviews.
Reviewer 2 Report
The manuscript by Pares-Casanova et al tests the hypothesis that there are differences in the shape of autopodial elements in the forelimbs of calves. They collected and radiographed specimens of 17 individuals all of the same breed that had not had managed foot care. They placed landmarks on each autopodial element and calculated shape differences using standard statistical methods.
The data presented in the manuscript are an anatomical description of distal limb elements in cattle. However, several conclusions in the abstract and discussion are not supported by evidence. The authors state in the abstract that asymmetries increase in directionality with weight bearing and are therefore 'functional' rather than developmental. This manuscript presents no evidence that asymmetries correlate with animal body weight, nor do I think one could separate weight from developmental progression as animals increase in mass concurrent with age. In the discussion, they elaborate on their inferred relationship to mechanical loading but again provide no supporting evidence. In order to state that asymmetries are a consequence of differences in mechanical loading, it would be important to, at the very least, demonstrate a correlation between element shape and load bearing on individual digits of each individual by biomechanical analyses. To make this conclusion even stronger, one should unload the digit to test the hypothesis that load bearing is required to manifest asymmetries.
Author Response
ll manuscript has been deeply modified and new references have been added. Conclusions realted to weight bearing and loadings have been deleted and conclusions are now in total agreement with “normal” anatomical hoof constitution. This is why some of Reviewer's comments are not discussed here. PLS analysis has been deleted (as we considered it added nothing to the research) and table 1 has been remodelled. Three figures have been added and the rest have been modified according to general reviews.
Reviewer 3 Report
Asymmetries of forelimb digits of young cattle
Abstract
Line 8 – ‘The lateral claw tends to be larger in the medial claw’ can you provide a references for this statement? Check author guidelines re refs in abstracts.
Line 12 - Why only forelimbs? How old are the calves? Alternatively, what is their weight?
Line 13 - meat not meet
Line 15 - Through line 16, the English needs some attention in this section as the meaning of the sentence is not clear to the first time reader.
Line 16 - it says for ‘both forelimbs it was observed’ this reads like just two limbs; do you mean for right and left limbs?
Line 17 – Your methodology will need to make sure it's really clear on how the limbs were positioned for radiography so that it's a repeatable.
Lines 17-18 - Again there is a little bit of difficulty understanding the intended meaning of the sentence due to the sentence structure.
Abstract – irregularities of font size between sentences in some areas.
Introduction:
General – referencing is not formatted in the number style requested by the author guidelines.
Flow within the introduction could be improved with attention to the written English and care that references are positioned correctly to support the statements made. Even for an Author/Date style the multiple listings are not polished, containing multiple brackets, which makes it hard to decide which statement the references are linked to.
Line 25 - When you refer to ‘between sides’ do you mean left and right sides of the animal, left and right sides of the limb, or the axial versus abaxial sides of a claw? Try and be really precise so there is no ambiguity for the reader.
Line 28 - ‘between others’ would probably be better written ‘among others’ but are these words really needed in this sentence?
Line 29 - FA – I would prefer to see this written in full at the start of a sentence, it is so much easier for the first time reader. In fact, longhand is easier than abbreviations unless you are close to the subject. You are discussing the application of what for many readers will be a new technique, so I would tend to write things in full unless you are really strapped for words. It saves the reader having to dip out and check what the shorthand meant again.
Line 31 – Can you expand this line to explain your reasoning here?
Line 32 - DA – I would prefer to see this written in full at the start of a sentence.
Line 33 – ‘… consist difference between sides’ – consistent difference in what?
Lines 33-35 - The meaning is pretty vague here can you tighten this wording up so it is more precise and streamlined.
Line 36 - The Cattle metacarpus…
Line 39 - three phalanges – if going to talk to this name them as proximal middle and distal....not sre two refs are needed here.
Lines 39-40 - I’m not sure this sentence has your intended meaning please check it over. Some linking words from one sentence to the next would help take the reader with your thought processes.
Line 42- (Mülling & Greenough, 2008) Great authors! However, this is not a peer-reviewed source can you find a peer reviewed reference for this statement? There is a fair bit of work done in this area- much presented at the International Ruminant Lameness conference (IRL) but then progressed to publication ?
Lines 43-44 – again you should be able to use a more recent source from the literature here – either IRL proceedings or preferably a peer-reviewed source, rather thanan older textbook. There has been a fair amount of work done and presented on the assymetry of the bones in this region (Nuss).
Lines 46-49 – Streamline two sentences to one only need reference once. You cite some interesting work from outside the usual literature for this topic try and flag to the reader the context of the work – e.g. Kubicka et al. (2016) is a human study based on a medieval population! Consider a brief overveiew of cartesian coordinates here, provide scaffold for the reader.
Line 54 - GM – I would prefer to see this written in full at the start of a sentence
Line 56 – I still wondering if the radiographs were taken live (having read right through I know they are not but I think you could make this clearer earlier on). ‘…thus, the present study being a pioneering one.’ This is not great English.
Line 57 – Need more detail here if at all possible - the age of the animals comes to mind again, how young is young? How long have they been turned out in the extensive system? Extensive covers so many systems can you narrow it down a bit? foot care not ‘feet care’.
Line 58 – ‘it is expected’
As above in the abstract there are irregularities of font size between sentences in some areas of the introduction.
Materials and methods
Line 61 – OK so these are limbs collected post-mortem – can you describe collection and preservation of limb ID – how did you know which limb was which? ‘Randomly’ - was it actually random? If so how did you achieve this randomisation? Or is this in fact a convenience sample from a local slaughterhouse? If so just tell us it is…
Line 63 - clinically sound and without apparent lameness – these two are the same thing.
Line 64 – How do you know there was no foot trimming of the animals? If you know this much detail surely you have age or live weights too which would enhance the content of this study.
Line 65 - Same thing for the sexes too – was there a mix of male and female animals? This background information informs the consideration of your reported findings.
Line 67 – Provide a brief outline of the digital radiography set up used (specifications and manufacturer).
Line 68 – If preparation of the foot is ‘essential’ then your methodology needs to include what that preparation involved – otherwise it is not repeatable.
Line 69 – an image or diagram of your set up would be good here. It would also be useful to illustrate where you centred your beam on the fetlock. (Fetlock is a colloquial term for the metacarpophalangeal joint).
Line 71 – why couldn’t the plate contact the limb? You need to define the landmarks you used in order for this to be repeatable. At some point in the paper you should expand on why these were the landmarks chosen.
Line 74 – are the captured images the radiographic images?
Line 78 – can you give examples of this technique being used in other peer reviewed publications?
Can you expand on how the configurations were compared?
Line 80 – Klingenberg (2015) is a MorphoJ reference you should mention the use of this software in your methodology. I'm sure you know what you mean but - why did you need a proxy for size? What size are you talking about? Define centroid size, for your mark-up scheme...
Figure 1- You need to include the numbers referred to in the text on this image or in the legend so that the reader can interpret the image in the light of the text. There is no left/right marker on the provided image – how did you know which limb was which? How did you minimise rotation of the severed limb and standardise the radiographic set up as much as possible to eliminate variation?
Legend speaks to natural stand not natural stance..? How was this achieved? 7 should be seven.
Statistical analysis
Lines 87-89 - Take care when explaining this to take the first time reader with you.
Line 90 - Your first mention of medial and lateral – are you referring to limbs or claws? Be precise to remove ambiguity. Make sure the nomenclature you use is clear and consistent throughout the manuscript.
Line 93 - FA – I would prefer to see this written in full at the start of a sentence
Line 95- help the reader by starting a new para and introducing some white space at ‘A canonical Variate…’
Line 96 – Make it really clear… ‘series’- medial and lateral? ‘each side’ - left and right?
Line 97 – I had to look up what CS was again – consider writing long hand…
Lines 100-101 – ‘The PLS test enables analysis of covariance between two sets of variables using the 100 fewest dimensions possible << insert a supporting reference >>. PLS – in full at the start of a sentence…
Line 102 – so morphoj was the stats not the measurements?
Line 103 – ‘Levels of statistical significance were computed by permutation tests’ – which tests?
Overall the methodolgy needs to be reported more fully with scafffolding for the reader.
Results
Present your descriptive data first ...
- Range of values weight, age…
- Were images obtained for all limbs?
Lines 105-107 – longwinded way to report the correlation obtained.
Line 109 - that not than?
Line 114 – why p <0.005 and not p < 0.0001 as in table 1?
Line 116 – Figure 2 not table 2.
Line 117 - All differences in the acropodial shape were significantly different – report the size of the effect for the reader. CV1 - new abbreviation not defined... (and at the start of a sentence).
Line 118 – percentages reported to two decimal places seems very precise for just 34 limbs. Is your model really this accurate?
Line 119- expand on content to incorporate what the groups are, this will reduce the cognitive load for the reader.
Lines 119-120 – Review the English structure. You need a few more words in here.
Table 1 – legend - left and right forlimbs of... You have n=30 here – what happened to the other 4 limbs? Define F and P – the table needs to stand alone and be fully interpretable without reference to the text.
Figure 2 - The image is a bit blurry can it be made clearer? Legend – include the colours here – if it is going to be reproduced in colour in the journal…again two decimal places in the percentages reported. I am looking for the features described as being in figure three in the text but there is no explanation in the way of labels or legend. label lateral and medial claws and the phalanges - help the reader interpret your diagram - you think this is a good method - sell it to us!
Line 137- in full at start of a sentence.
Lines 137-139 – You need to use more words here more words to aid the explanation. 4 should be four and 7 should be seven in this section.
Line 141 – What does this sentence mean? Do you mean significant regression? In which case the p value given does not report the sentence... You need to report the regression equation the R2 and your confidence intervals here...
Line 148 - good to know how calves were housed if extensive system not much pressure compared to dairy breeds..
Discussion
Line 144 – young – how old / or at least what live weight.
Line 148 – this is where it would be good to know how calves were raised and again how young is young? If they are in a truly extensive system there would be very little pressure on lateralized behaviour compared to dairy breeds. What distribution of sidedness would be expected? How does limb size vary with sidedness? Can you expand on this?
Line 150 - It would be nice to see these results discussed within the existing cattle literature... Nüske looked at dimensions, bone densities in relation to growth in young calves. There was a second paper too (sorry I don't have the referenc eto hand for it). These might help provide some context even though you have focussed on a more proximal region of the limb.
- Nüske, S., Scholz, A.M., Förster, M., 2003. Studies on the growth and the development of the Claw capsule in new born calves of different breeding lines using linear measurements. Archiv fur Tierzucht 46, 547-557.
Line 153 – Again there is publiseded work on this which could strengthen your discussion.
Line 154 – what are the othere explanations? Can you find some supporting references for them?
Line 155 – Define what you mean by size here?
Line 157 – There is literature looking at the volume of claws and some speak to the size of animal too. Here are a couple of references.
- Phillips, C.J.C., Patterson, S.J., Dewi, I.A., 1996. Volume assessment of the bovine hoof. Research in Veterinary Science 61, 125-128.
- Clark, C.R., Petrie, L., Waldner, C., Wendell, A., 2004. Characteristics of the bovine claw associated with the presence of vertical fissures (sandcracks). Canadian Veterinary Journal 45, 585-593.
Line 160 – can you offer more or a potential explanation for this finding?
Line 162 – I agree did you have a mix of sexes you don’t tell us – I suspect you would need more back ground infomation and more limbs! Again you refer to more aged but have not told us how old your population was.
Line 163 – there are quite a few kinematic and force plate studies out there which may enable you to discuss this more fully.
- Meyer, S.W., Weishaupt, M.A., Nuss, K.A., 2007. Gait pattern of heifers before and after claw trimming: A high-speed cinematographic study on a treadmill. Journal of dairy Science 90, 670-676.
- Schmid, T., Weishaupt, M.A., Meyer, S.W., Waldern, N., Peinen, K.v., Nuss, K., 2009. High-speed cinematographic evaluation of claw-ground contact pattern of lactating cows. Veterinary Journal 181, 151-157.
- van der Tol, P.P.J., Metz, J.H.M., Noordhuizen-Stassen, E.N., Back, W., Braam, C.R., Weijs, W.A., 2003. The vertical ground reaction force and the pressure distribution on the claws of dairy cows while walking on a flat substrate. Journal of Dairy Science 86, 2875-2883.
I think the application of this technique could be of interest to people working in this field. However, I would really like to see the study and it's results discussed within the context of the work which has been previously been published with respect to Cattle. There are quite a lot of published studies that could help you suggest how could this technique be used to further knowledge in this field.
Author Response
All manuscript has been deeply modified and new references have been added. Conclusions realted to weight bearing and loadings have been deleted and conclusions are now in total agreement with “normal” anatomical hoof constitution. This is why some of Reviewer's comments are not discussed here. PLS analysis has been deleted (as we considered it added nothing to the research) and table 1 has been remodelled. Three figures have been added and the rest have been modified according to general reviews.
Line 8 – ‘The lateral claw tends to be larger in the medial claw’ can you provide a references for this statement? Check author guidelines re refs in abstracts.
A reference has been added.
Line 12 - Why only forelimbs? How old are the calves? Alternatively, what is their weight?
Age and carcass weight of animals have been included. Justification for forelimbs (less prone to changes to substrates) too.
Line 13 - meat not meet Corrected, my mistake.
Line 15 - Through line 16, the English needs some attention in this section as the meaning of the sentence is not clear to the first time reader.
I have tried to rewrite all the text in a more “light” way.
Line 16 - it says for ‘both forelimbs it was observed’ this reads like just two limbs; do you mean for right and left limbs? Corrected.
Line 17 – Your methodology will need to make sure it's really clear on how the limbs were positioned for radiography so that it's a repeatable.
Some key points have been added: “with all the solar hoof surface on a block”, “the cassette (…) parallel as possible to the metapode”.
Lines 17-18 - Again there is a little bit of difficulty understanding the intended meaning of the sentence due to the sentence structure. Paragraph modified.
Abstract – irregularities of font size between sentences in some areas. Revised.
Introduction:
General – referencing is not formatted in the number style requested by the author guidelines.
Flow within the introduction could be improved with attention to the written English and care that references are positioned correctly to support the statements made. Even for an Author/Date style the multiple listings are not polished, containing multiple brackets, which makes it hard to decide which statement the references are linked to.
Line 25 - When you refer to ‘between sides’ do you mean left and right sides of the animal, left and right sides of the limb, or the axial versus abaxial sides of a claw? Try and be really precise so there is no ambiguity for the reader. Corrected.
Line 28 - ‘between others’ would probably be better written ‘among others’ but are these words really needed in this sentence? Corrected, as now it is only for bilateral asymmetries.
Line 29 - FA – I would prefer to see this written in full at the start of a sentence, it is so much easier for the first time reader. In fact, longhand is easier than abbreviations unless you are close to the subject. You are discussing the application of what for many readers will be a new technique, so I would tend to write things in full unless you are really strapped for words. It saves the reader having to dip out and check what the shorthand meant again. Corrected. Thanks for this tip.
Line 31 – Can you expand this line to explain your reasoning here? ----
Line 32 - DA – I would prefer to see this written in full at the start of a sentence. Done.
Line 33 – ‘… consist difference between sides’ – consistent difference in what? Corrected: “consistent form differences...”.
Lines 33-35 - The meaning is pretty vague here can you tighten this wording up so it is more precise and streamlined.
Line 36 - The Cattle metacarpus… Corrected.
Line 39 - three phalanges – if going to talk to this name them as proximal middle and distal....not sre two refs are needed here. All sentence has been modified.
Lines 39-40 - I’m not sure this sentence has your intended meaning please check it over. Some linking words from one sentence to the next would help take the reader with your thought processes. -----
Line 42- (Mülling & Greenough, 2008) Great authors! However, this is not a peer-reviewed source can you find a peer reviewed reference for this statement? There is a fair bit of work done in this area- much presented at the International Ruminant Lameness conference (IRL) but then progressed to publication ?
Lines 43-44 – again you should be able to use a more recent source from the literature here – either IRL proceedings or preferably a peer-reviewed source, rather thanan older textbook. There has been a fair amount of work done and presented on the assymetry of the bones in this region (Nuss).
Lines 46-49 – Streamline two sentences to one only need reference once. You cite some interesting work from outside the usual literature for this topic try and flag to the reader the context of the work – e.g. Kubicka et al. (2016) is a human study based on a medieval population! Consider a brief overveiew of cartesian coordinates here, provide scaffold for the reader.
Line 54 - GM – I would prefer to see this written in full at the start of a sentence
Line 56 – I still wondering if the radiographs were taken live (having read right through I know they are not but I think you could make this clearer earlier on). ‘…thus, the present study being a pioneering one.’ This is not great English.
Corrected: “To ouw knowledge, radiographic geometric morphometric analyses of digits have not previously been used for cattle”.
Line 57 – Need more detail here if at all possible - the age of the animals comes to mind again, how young is young? How long have they been turned out in the extensive system? Extensive covers so many systems can you narrow it down a bit? foot care not ‘feet care’. We have supressed this sentence, too vague.
Line 58 – ‘it is expected’ Corrected, thanks.
As above in the abstract there are irregularities of font size between sentences in some areas of the introduction.
Materials and methods
Line 61 – OK so these are limbs collected post-mortem – can you describe collection and preservation of limb ID – how did you know which limb was which? ‘Randomly’ - was it actually random? If so how did you achieve this randomisation? Or is this in fact a convenience sample from a local slaughterhouse? If so just tell us it is… Added “and labelled”. “Randomly” was supressed.
Line 63 - clinically sound and without apparent lameness – these two are the same thing. A redundancy, corrected.
Line 64 – How do you know there was no foot trimming of the animals? If you know this much detail surely you have age or live weights too which would enhance the content of this study. It was logical by visual inspection of each hoof. Anyway, trimming is never done on these animals.
Line 65 - Same thing for the sexes too – was there a mix of male and female animals? This background information informs the consideration of your reported findings. It is stated we had no individual information (due to abattoir's convenience).
Line 67 – Provide a brief outline of the digital radiography set up used (specifications and manufacturer). Clarified: a Potro DR machine..
Line 68 – If preparation of the foot is ‘essential’ then your methodology needs to include what that preparation involved – otherwise it is not repeatable. Clarified.
Line 69 – an image or diagram of your set up would be good here. It would also be useful to illustrate where you centred your beam on the fetlock. (Fetlock is a colloquial term for the metacarpophalangeal joint). Corrected.
Line 71 – why couldn’t the plate contact the limb? You need to define the landmarks you used in order for this to be repeatable. At some point in the paper you should expand on why these were the landmarks chosen
Paragraph modified. A positioning image has been added to figure 1.
Line 74 – are the captured images the radiographic images? Yes, I think it is clear.
Line 78 – can you give examples of this technique being used in other peer reviewed publications? Can you expand on how the configurations were compared?
Line 80 – Klingenberg (2015) is a MorphoJ reference you should mention the use of this software in your methodology. I'm sure you know what you mean but - why did you need a proxy for size? What size are you talking about? Define centroid size, for your mark-up scheme... It is needed for allometric analysis.
Figure 1- You need to include the numbers referred to in the text on this image or in the legend so that the reader can interpret the image in the light of the text. There is no left/right marker on the provided image – how did you know which limb was which? How did you minimise rotation of the severed limb and standardise the radiographic set up as much as possible to eliminate variation?
Legend speaks to natural stand not natural stance..? How was this achieved? 7 should be seven.
Statistical analysis
Lines 87-89 - Take care when explaining this to take the first time reader with you. In our opinion explanations have been improved.
Line 90 - Your first mention of medial and lateral – are you referring to limbs or claws? Be precise to remove ambiguity. Make sure the nomenclature you use is clear and consistent throughout the manuscript. In our opinion explanations have been improved.
Line 93 - FA – I would prefer to see this written in full at the start of a sentence Line 95- help the reader by starting a new para and introducing some white space at ‘A canonical Variate…’
Line 96 – Make it really clear… ‘series’- medial and lateral? ‘each side’ - left and right?
Line 97 – I had to look up what CS was again – consider writing long hand…
Lines 100-101 – ‘The PLS test enables analysis of covariance between two sets of variables using the 100 fewest dimensions possible << insert a supporting reference >>. PLS – in full at the start of a sentence…
This analysis has been deleted, as we considere it is confused and adds nothing to the main conclusions.
Line 102 – so morphoj was the stats not the measurements?
MorphoJ does not work with measurements but with procrustes coordinates!
Line 103 – ‘Levels of statistical significance were computed by permutation tests’ – which tests? Overall the methodolgy needs to be reported more fully with scafffolding for the reader. Improved.
Results
Present your descriptive data first ...
Range of values weight, age…
Were images obtained for all limbs?
Lines 105-107 – longwinded way to report the correlation obtained.
Line 109 - that not than?
Line 114 – why p <0.005 and not p < 0.0001 as in table 1? Content in the table is now different.
Line 116 – Figure 2 not table 2. Corrected.
Line 117 - All differences in the acropodial shape were significantly different – report the size of the effect for the reader. CV1 - new abbreviation not defined... (and at the start of a sentence).
Line 118 – percentages reported to two decimal places seems very precise for just 34 limbs. Is your model really this accurate? Yes.
Line 119- expand on content to incorporate what the groups are, this will reduce the cognitive load for the reader.
Lines 119-120 – Review the English structure. You need a few more words in here.
Table 1 – legend - left and right forlimbs of... You have n=30 here – what happened to the other 4 limbs? Define F and P – the table needs to stand alone and be fully interpretable without reference to the text. It was a mistake.
Figure 2 - The image is a bit blurry can it be made clearer? Legend – include the colours here – if it is going to be reproduced in colour in the journal…again two decimal places in the percentages reported. I am looking for the features described as being in figure three in the text but there is no explanation in the way of labels or legend. label lateral and medial claws and the phalanges - help the reader interpret your diagram - you think this is a good method - sell it to us! Figures have been modified and some added. In our opinion we have gained clarity.
Line 137- in full at start of a sentence.
Lines 137-139 – You need to use more words here more words to aid the explanation. 4 should be four and 7 should be seven in this section.
Line 141 – What does this sentence mean? Do you mean significant regression? In which case the p value given does not report the sentence... You need to report the regression equation the R2 and your confidence intervals here...
Line 148 - good to know how calves were housed if extensive system not much pressure compared to dairy breeds..
Discussion
Line 144 – young – how old / or at least what live weight. Line 148 – this is where it would be good to know how calves were raised and again how young is young? If they are in a truly extensive system there would be very little pressure on lateralized behaviour compared to dairy breeds. What distribution of sidedness would be expected? How does limb size vary with sidedness? Can you expand on this?
Line 150 - It would be nice to see these results discussed within the existing cattle literature... Nüske looked at dimensions, bone densities in relation to growth in young calves. There was a second paper too (sorry I don't have the referenc eto hand for it). These might help provide some context even though you have focussed on a more proximal region of the limb. Nüske, S., Scholz, A.M., Förster, M., 2003. Studies on the growth and the development of the Claw capsule in new born calves of different breeding lines using linear measurements. Archiv fur Tierzucht 46, 547-557.
Line 153 – Again there is publiseded work on this which could strengthen your discussion.
Line 154 – what are the othere explanations? Can you find some supporting references for them?
Line 155 – Define what you mean by size here?
Line 157 – There is literature looking at the volume of claws and some speak to the size of animal too. Here are a couple of references.
Phillips, C.J.C., Patterson, S.J., Dewi, I.A., 1996. Volume assessment of the bovine hoof. Research in Veterinary Science 61, 125-128.
Clark, C.R., Petrie, L., Waldner, C., Wendell, A., 2004. Characteristics of the bovine claw associated with the presence of vertical fissures (sandcracks). Canadian Veterinary Journal 45, 585-593.
First article has been introduced into the text. Second one is very interesting but lacks of information for our research.
Line 160 – can you offer more or a potential explanation for this finding?
Line 162 – I agree did you have a mix of sexes you don’t tell us – I suspect you would need more back ground infomation and more limbs! Again you refer to more aged but have not told us how old your population was. Age of animals has been introduced in the Materials & Methods section.
Line 163 – there are quite a few kinematic and force plate studies out there which may enable you to discuss this more fully.
Meyer, S.W., Weishaupt, M.A., Nuss, K.A., 2007. Gait pattern of heifers before and after claw trimming: A high-speed cinematographic study on a treadmill. Journal of dairy Science 90, 670-676.
Schmid, T., Weishaupt, M.A., Meyer, S.W., Waldern, N., Peinen, K.v., Nuss, K., 2009. High-speed cinematographic evaluation of claw-ground contact pattern of lactating cows. Veterinary Journal 181, 151-157.
van der Tol, P.P.J., Metz, J.H.M., Noordhuizen-Stassen, E.N., Back, W., Braam, C.R., Weijs, W.A., 2003. The vertical ground reaction force and the pressure distribution on the claws of dairy cows while walking on a flat substrate. Journal of Dairy Science 86, 2875-2883.
I think the application of this technique could be of interest to people working in this field. However, I would really like to see the study and it's results discussed within the context of the work which has been previously been published with respect to Cattle. There are quite a lot of published studies that could help you suggest how could this technique be used to further knowledge in this field.
Discussion is not centered on weight bearing and loading, as no functional data were available for our studied sample.
Round 2
Reviewer 2 Report
The revised version of this manuscript has been improved by adjusting the writing to remove unsupported interpretations that locomotion changes shape. I am now, however, confused by the new 'bending energy' data presented in Figure 6, as the methods do not adequately relate these data to the radiographs. Please improve the methods and the description of these data in the results.
Other minor points.
Line 47: "Researches analyzing" - change to "Analyses of"
Line 50: "no equal" - change to "not equal"
Line 53: "minor roler" - change to "lesser role"
Line 59: "in a haphazard manner" - change to "at random"
Author Response
All minor points have been modified according to your comments.
"Bending energy" concept has also been clarified (lines 105-106).

Reviewer 3 Report
Comments are included in the attached file.

Author Response
All your comments have been introduced. Second reviewer's comments appear also highlighted.
